# Cerebral Organoids Maintain the Expression of Neural Stem Cell-Associated Glycoepitopes and Extracellular Matrix

**DOI:** 10.3390/cells11050760

**Published:** 2022-02-22

**Authors:** Lars Roll, Katrin Lessmann, Oliver Brüstle, Andreas Faissner

**Affiliations:** 1Department of Cell Morphology and Molecular Neurobiology, Ruhr University Bochum, 44780 Bochum, Germany; 2Institute of Reconstructive Neurobiology, Medical Faculty & University Hospital Bonn, University of Bonn, 53127 Bonn, Germany; brustle@uni-bonn.de

**Keywords:** cerebral organoid, chondroitin sulfate, DSD-1 epitope, extracellular matrix, glycoepitope, LewisX epitope, neural stem cell, PTPRZ1, radial glia, Tenascin C

## Abstract

During development, the nervous system with its highly specialized cell types forms from a pool of relatively uniform stem cells. This orchestrated process requires tight regulation. The extracellular matrix (ECM) is a complex network rich in signaling molecules, and therefore, of interest in this context. Distinct carbohydrate structures, bound to ECM molecules like Tenascin C (TNC), are associated with neural stem/progenitor cells. We have analyzed the expression patterns of the LewisX (LeX) trisaccharide motif and of the sulfation-dependent DSD-1 chondroitin sulfate glycosaminoglycan epitope in human cerebral organoids, a 3D model for early central nervous system (CNS) development, immunohistochemically. In early organoids we observed distinct expression patterns of the glycoepitopes, associated with rosette-like structures that resemble the neural tube in vitro: Terminal LeX motifs, recognized by the monoclonal antibody (mAb) 487^LeX^, were enriched in the lumen and at the outer border of neural rosettes. In contrast, internal LeX motif repeats detected with mAb 5750^LeX^ were concentrated near the lumen. The DSD-1 epitope, labeled with mAb 473HD, was detectable at rosette borders and in adjacent cells. The epitope expression was maintained in older organoids but appeared more diffuse. The differential glycoepitope expression suggests a specific function in the developing human CNS.

## 1. Introduction

The development of the nervous system starts with the induction of the neuroectoderm, followed by the formation of the neural tube by cells with a radial morphology. These cells, first called neuroepithelial cells and later radial glia, represent neural stem cells that will give rise to all neurons, astrocytes, and oligodendrocytes of the adult nervous system. Emanating from the single-layered neural tube, differentiation and migration processes lead to the formation of complex layers, as seen in the neocortex and other neural tissues [1,2].

All these events require tight regulation of processes, such as proliferation, migration and differentiation. In combination with intrinsic programs of the cells, extracellular signals are indispensable for proper nervous system development. In this context, the extracellular matrix (ECM) is of utmost interest [3]. The ECM, a complex network of interacting molecules and consisting of ca. 300 proteins as part of the core matrisome, contains a multitude of signaling molecules and interacts with additional factors like cytokines and growth factors, which are eventually presented to the cells and influence the cells’ behavior [4]. The interaction of molecules within the ECM is a crucial aspect of its function and different modifications exist to fine-tune such interactions. Glycosylation, namely the attachment of carbohydrates to ECM molecules, is one type of modification that can alter the interaction of ECM molecules and thereby modulate the effect on cells that are in contact with this ECM [5,6].

Several well-defined carbohydrate structures have been described so far that show an astonishing degree of specificity regarding their expression profile and also function. An important structure is the LewisX (LeX) motif, which consists of galactose 1-4-linked and fucose1-3-linked to N-acetylglucosamine [Galβ1-4(Fucα1-3)GlcNAc] and is associated with neural stem cells [7,8]. Monoclonal antibodies (mAbs) that label the LeX motif depending on its position within a carbohydrate are available. mAb 487^LeX^ only binds to the LeX motif if it is at a terminal position of a carbohydrate structure, whereas binding of mAb 5750^LeX^ requires the presence of internal repeats of the LeX trisaccharide within a carbohydrate. Using these mAbs, distinct subpopulations of neural stem cells from the rodent brain could be distinguished in the past [9]. Another important carbohydrate motif is the Dermatan sulfate-dependent 1 (DSD-1) epitope, a sulfation-dependent chondroitin sulfate structure that is recognized by mAb 473HD [10]. It has been shown to be essential for proper mouse neural stem cell function [11]. Inhibited sulfation, in general, affects neural stem cell proliferation and differentiation [12,13]. We focus our analysis on two potential LeX carriers, namely Tenascin C (TNC) and Protein tyrosine phosphatase receptor type Z1 (PTPRZ1). Both molecules interact with each other [14,15]. TNC is an important component of the embryonic and adult neural stem cell niche [16,17,18]. PTPRZ1 isoforms, of which two can carry the DSD-1 epitope, are also expressed in the immature nervous system, where they regulate neurite outgrowth and, as carriers of the DSD-1 epitope, neural stem cell behavior [11,19].

In the light of diseases and potential therapeutic approaches, a better understanding of the processes in the developing nervous system is desired. Although many features are shared between species, models to study the human nervous system are required, as the human brain with its gyrified neocortex is undoubtedly different compared to the brain of the mouse and other popular model organisms. Cell characteristics and cell functions, but also pathomechanisms can differ between species, as shown for Alzheimer’s disease [20,21,22]. These and other findings emphasize the need for studies performed with human cells to verify and complement the results obtained in animal models. Human induced pluripotent stem cells (hiPSCs) can be reprogrammed from adult somatic cells, which makes them easily available and avoids the destruction of a human embryo. The technique was established by Yamanaka and colleagues, first with mice and later also with human cells [23,24]. iPSCs are a valuable tool to generate the different cell types of the body and protocols have been developed that allow the stepwise differentiation into specific cell types [25]. Approaches that are based on two-dimensional cell cultures have the advantage of well-defined and controllable conditions. In this system, we have analyzed the expression of the LeX and DSD-1 epitopes and found a neural stem cell-associated expression pattern [26]. Three-dimensional cell cultures provide a more complex, but at the same time less defined environment for the cells. Such systems can be used to study cell migration, differentiation, alterations in gene expression signatures and more. Aggregates formed by neural stem cells are referred to as neurospheres [27], but there is a wide range of applications, including cancer research and models for drug resistance and toxicity studies [28,29,30,31,32].

Organoids are a relatively new approach. Here, the self-organizing capacity of cells is exploited to generate complex three-dimensional organ-like cell aggregates, with the advantage of a more in vivo-like situation for the cells compared to an adherent culture on a flat artificial surface, as mentioned above. This technique allows the generation of cell aggregates that resemble a wide range of tissues of the body, including the developing brain. In this case, the aggregates are called cerebral organoids [33,34]. Manifold conditions of the nervous system can be and have already been addressed with this organoid approach. For example, microcephalic and lissencephalic malformations were modeled, as well as aspects of the neurodegenerative Alzheimer’s disease [33,35,36]. Additionally, the behavior of glioma cells can be analyzed in cerebral organoids [37].

In the present study, we analyzed the expression of the LeX and DSD-1 glycoepitopes, as well as the expression of the two potential carrier molecules PTPRZ1 and TNC in hiPSC-derived cerebral organoids. The spatiotemporal expression patterns and the distribution of the molecules within the organoids might point to their function in this 3D model for human CNS development.

## 2. Materials and Methods

### 2.1. Cell Culture of Human Induced Pluripotent Stem Cells (hiPSCs)

The protocol for the 3D cerebral organoid culture is based on the technique established by Lancaster and Knoblich, with small modifications [38]. A detailed description is available as supplemental material (Appendix A and Figure A1 in the Appendix B), detailed information about the materials, including catalog numbers and buffer compositions, is provided in Table A1, Table A2, Table A3, Table A4, Table A5, Table A6 and Table A7 in the Appendix C. In brief, the following approach was used:

The hiPS cell line iLB-C-31f-r1^3^ was established at the Institute of Reconstructive Neurobiology, Life & Brain Center, University of Bonn [26]. hiPSCs were cultured feeder-free on 6-well plates (Nunc/Thermo, Roskilde, Denmark) coated with Matrigel (Corning, Wiesbaden, Germany #354234). The cells were thawed and resuspended in mTeSR^TM^1 medium (Stemcell Technologies, Cologne, Germany) supplemented with Rho-associated protein kinase (ROCK) inhibitor (1:1000; Merck, Darmstadt, Germany) and kept at 37 °C and 5% CO_2_ for 4 days in vitro (DIV). The medium was replaced every 24 h. After 4 DIV, the hiPSC colonies were ca. 80% confluent and could be used to perform the next steps.

#### 2.1.1. Generation of Cells with Neural Identity

The aim was to induce a neuroectodermal identity in cells generated in human Embryoid Bodies (hEBs). Therefore, hEBs were generated in a first step before they underwent neural induction.

#### 2.1.2. hEB Formation (0–6 DIV)

The hiPSC colonies were enzymatically digested with Accutase (Sigma-Aldrich, Steinheim, Germany) and detached from the surface. The cells were resuspended in mTeSR^TM^1 with ROCK inhibitor, counted and 25 μL-drops of a cell suspension with 9000 cells each were placed carefully on the inverted lid of a Petri dish (VWR, Darmstadt, Germany). The dish was filled with PBS to reduce evaporation, the lid was closed and after 3 days of incubation as “hanging drops”, the generated hEBs were collected and transferred into separate wells of a 96-well plate with U-shaped bottoms (Corning, Wiesbaden, Germany), filled with mTeSR^TM^1 plus ROCK inhibitor. Afterward, the hEBs were cultured for 3 days. Half of the medium was replaced after 2 DIV by a fresh medium w/o ROCK inhibitor. Alternatively to the abovementioned method, hEBs were generated in low attachment-96-well plates with U-shaped bottoms. Dissociated cells were transferred into the wells and incubated for 6 days. The medium was changed every other day, no ROCK inhibitor was added after the second medium change.

#### 2.1.3. Neural Induction (6–11 DIV)

hEBs that reached a diameter of ca. 500–650 μm after 6 days and had a smooth surface, were transferred into separate wells of a low attachment-24-well plate (Corning, Wiesbaden, Germany), filled with Neural Induction Medium. Incubation for 5 days at 37 °C and 5% CO_2_ followed. Medium was replaced after 48 h. 

#### 2.1.4. Transfer of Neuroepithelial Tissue into Matrigel (11 DIV)

To model the complex in vivo microenvironment in vitro and to allow a successful 3D expansion of the highly organized neuroepithelial structures, the neurally induced hEBs were embedded in the hydrated Matrigel (Corning, Wiesbaden, Germany #354230) [39]. To form three-dimensional Matrigel drops, a mold was prepared from Parafilm^®^ (Bemis, Neenah, WI, USA) and each neural aggregate was transferred into a dimple of the prepared Parafilm^®^. The excess medium was removed. Subsequently, 30 μL Matrigel was added to each aggregate, before the Matrigel started to solidify. The dish with the drops was incubated at 37 °C to induce proper polymerization of the Matrigel. After the 6 cm dish (Nunc/Thermo, Roskilde, Denmark) has been flooded with Organoid Differentiation Medium w/o vitamin A, the Matrigel-coated aggregates were released by inverting the Parafilm^®^ with sterile forceps and by careful shaking. Afterward, the culture was continued in the incubator.

#### 2.1.5. Static Organoid Culture (11–15 DIV)

The aggregates, embedded in Matrigel, were incubated for 4 days. After 48 h the medium in the 6 cm cell culture dish was completely removed and replaced by fresh Organoid Differentiation Medium w/o vitamin A.

#### 2.1.6. Dynamic Organoid Culture (15 DIV and Later)

After 4 days in the static cell culture, the old medium was replaced by Organoid Differentiation Medium containing vitamin A and the dish was placed on a shaker. The medium was changed every 3–4 days and the cells could be maintained for several months.

#### 2.1.7. Preparation of Cerebral Organoids for Cryosectioning

Cerebral organoids were fixed for 15 min at 4 °C with 4% (*w*/*v*) paraformaldehyde and washed with PBS. Subsequently, they were dehydrated with 30% (*w*/*v*) sucrose solution overnight at 4 °C. On the next day, the organoids were completely embedded in gelatin/sucrose solution (7.5% (*w*/*v*) gelatin and 10% (*w*/*v*) sucrose in PBS). After the gelatin was allowed to gel, small blocks of gelatin with the organoids inside were cut out. The blocks were frozen in a prepared bath of isopentane on dry ice at −30 to −50 °C and then stored at −80 °C; 14–20 μm thick slices were generated with a cryostat and transferred to glass slides (Menzel/Thermo, Braunschweig, Germany).

### 2.2. Immunohistochemistry and Fluorescence Microscopy

For immunohistochemical analyses, slices were washed with PBS and each slice was encircled with a barrier marker (Carl Roth, Karlsruhe, Germany). After blocking for 1 h with blocking buffer, primary antibodies (Table 1; diluted in blocking buffer) were incubated overnight at 4 °C. After three steps of washing with PBS, the appropriate secondary antibodies (Table 2; diluted in PBS, with 1% (*w*/*v*) BSA and 1% (*v*/*v*) goat serum (Dianova, Hamburg, Germany)) were incubated together with TO-PRO^®^-3 (1:500; Thermo Scientific, Waltham, MA, USA) for 1 h at RT. Finally, the samples were washed three times with PBS and covered with ImmuMount^®^ (Thermo Scientific, Waltham, MA, USA). Samples were documented with a fluorescence microscope (Axio Zoom.V16; Carl Zeiss, Jena, Germany) and a laser scanning microscope (LSM 510 META; Carl Zeiss, Jena, Germany). Maximum intensity projections were generated with the ZEN software (Carl Zeiss, Jena, Germany) from z-stacks of confocal images taken with the laser scanning microscope (LSM). Image processing was performed with ZEN software (Carl Zeiss, Jena, Germany), Photoshop and Illustrator software (Adobe Systems, San Jose, CA, USA) as well as with Inkscape.

We analyzed the glycoepitope expression patterns in at least three young organoids (up to five weeks after embedding in Matrigel) with the typical neural rosette structure and at least three late organoids (at least six weeks after embedding in Matrigel), respectively. To illustrate the expression patterns of the glycoepitopes, we generated fluorescence intensity profiles using the tool “Plot Profile” of FIJI/ImageJ.

## 3. Results

After neural induction of the human Embryoid Bodies according to the protocol of Lancaster et al. [38], the morphology of the cell aggregates changed dramatically. Small protrusions formed on the surface of the early cerebral organoid after 18 days in vitro (DIV) (Figure 1A). With ongoing maturation, the surface became smoother, as shown after 32 DIV and 95 DIV. Only a few, larger protrusions remained (Figure 1B,C). The protrusions were formed by neural stem cells with a radial morphology, which was confirmed by immunohistochemical staining for cell type-specific markers.

### 3.1. Cell Fate and Proliferation in Human Cerebral Organoids

To characterize the cells that were present in the cerebral organoids, we performed immunohistochemical stainings. The first aspect was the proliferation activity in early cerebral organoids after 18 DIV (Figure 2). KI-67, a marker for all proliferating cells, was expressed by the vast majority of all cells, as the comparison with nuclear TO-PRO-3 staining showed (Figure 2A). Phospho-Histone H3 (PH3), a marker that is restricted to late G2- and M-phase cells, was expressed only by a minority of cells. Here, it was cells within rosette-like structures, where the PH3 signals were primarily found at the apical side (Figure 2B). These rosette-shaped structures are typical of neural cell cultures and represent an in vitro counterpart of the neural tube, the structure of the early nervous system in vivo. It is formed by neural stem cells with radial morphology. Therefore, the strong expression of Nestin, the prototypical neural stem cell marker, in these structures confirmed the cell fate we expected. Additionally, the localization of PH3-positive cell somas only in direct vicinity to the apical side of the cells, is in line with the conditions found in the neural tube and reflects a mechanism called “interkinetic nuclear migration”.

In older cerebral organoids (Figure 3), the ratio of proliferating cells declined over time. This was shown by the comparison of KI-67 expression in cerebral organoids after 25 DIV (Figure 3A) with the situation after 53 DIV or even after 81 DIV (Figure 3B,C). In the older organoids, the remaining KI-67 signals were restricted to the rosette structures after 53 DIV, and after 81 DIV, when the rosettes had vanished, they were only found in the outer regions of the organoid. Like Nestin, the glutamate aspartate transporter (GLAST; synonym: EAAT1), a marker for the radial glia type of neural stem cells, was strongly expressed in the neural rosette structures in the early organoids, whereas the signal appeared more diffuse after 81 DIV.

In the next step, we characterized the cell fate in the cerebral organoids over time in more detail by means of cell type-specific markers.

### 3.2. Cell Types in the Cerebral Organoids

To address the question of the different neural cell types in more detail, we analyzed the expression patterns of the cell type-specific markers SRY-box transcription factor 2 (SOX2), βIII-Tubulin (also TUBB3), Oligodendrocyte transcription factor 2 (OLIG2) and Glial fibrillary acidic protein (GFAP) in early and later cerebral organoids (Figure A2 and Figure A3 in the Appendix B). SOX2, which labels neuroepithelial cells and radial glia, was expressed by the majority of cells, with a clear association to cells within the neural rosettes after 25 DIV (Figure A2B). The neuronal marker βIII-Tubulin was virtually undetectable after 18 DIV, but appeared after 25 DIV in cells outside the neural rosette-like regions (Figure A2A,B). The progenitor and oligodendroglial marker OLIG2 could not be detected after 18 and 25 DIV and also the astrocyte marker GFAP was only weakly expressed in early cerebral organoids. It was restricted to very few regions within the aggregates (Figure A2C,D).

The situation changed with the ongoing maturation of the organoids (Figure A3). Expression of SOX2 was limited to neural rosette-like regions after 53 DIV and to outer regions after 81 DIV when rosettes structures vanished. βIII-Tubulin as a neuronal marker showed an inverted image compared to that of SOX2. Neurons were labeled outside the neural rosette structures after 53 DIV, also after 81 DIV, the βIII-Tubulin-positive areas were poor in SOX2-expressing cells and vice versa (Figure A3A,B). OLIG2, a marker for progenitors and oligodendroglial cells, could not be detected, even in older organoids (Figure A3C,D). In contrast, GFAP-positive astrocytes were detected in the outer regions of the organoid. Their distribution was not even, instead large regions were free of GFAP-expressing astrocytes.

### 3.3. Expression of LewisX (LeX) Motifs and Potential Carriers in Cerebral Organoids

After the different cell types had been characterized, double staining for the LewisX (LeX) motif and for two potential LeX carrier molecules, namely Protein tyrosine phosphatase receptor type Z1 (PTPRZ1) and Tenascin C (TNC), was performed. TNC is one of several ECM molecules that can be decorated with LeX epitopes. PTPRZ1 is also a potential LeX carrier and the only confirmed carrier of the DSD-1 chondroitin sulfate epitope. Signals of the monoclonal antibody (mAb) 487^LeX^, which binds to terminal LeX motifs, were most prominent in the lumen and at the outer border of neural rosette structures after 32 DIV (Figure 4A and Figure A4A). PTPRZ1 showed a similar, but not identical staining pattern compared to the 487^LeX^ signals. The most intense staining was detectable on cells within the neural rosette structures.

Later, after 81 DIV, the signals of 487^LeX^ and for PTPRZ1 appeared more diffuse and showed no clear co-localization (Figure 4B and Figure A4B). mAb 5750^LeX^, which binds to internal LeX motif repeats, labeled the lumen of neural rosettes after 32 DIV. The outer borders of these structures were only weakly stained. TNC was mainly labeled at the rosette borders (Figure 4C). After 81 DIV, signals for mAb 5750^LeX^ and for TNC could still be detected in the organoid. In these late organoids, regions with intensely labeled cells could be observed, but at the same time, regions with very faint signals were present (Figure 4D). The strong signals in the core of late organoids indicate necrosis due to insufficient oxygen supply in the cell culture, and therefore, cannot be considered specific.

As described, the analyzed neural stem cell-related carbohydrate motifs were associated with the neural tube-resembling rosette structures. To analyze how far the Nestin-positive cells were indeed related to the LeX structures and the potential carriers TNC and PTPRZ1, we performed double immunostainings for Nestin combined with these matrix molecules (Figure 5). After 18 DIV, Nestin signals were most prominent in the neural rosettes and 487^LeX^ signals were detectable at the lumen and at the outer border of these neural rosettes. In contrast, 5750^LeX^ signals were most intense in the lumen of the rosettes (Figure 5A,B and Figure A4). TNC, a potential LeX carrier, was also detectable on cells of the neural rosettes, although it appeared weaker compared to the strong signals seen for PTPRZ1 (Figure 5C,D).

After 81 DIV, the rosette-like structures disappeared and the staining patterns of the molecules changed accordingly. Nestin-positive cells were distributed throughout the organoid and the signals for the LeX motifs, for TNC and for PTPRZ1 did not show a clear expression pattern (Figure 5E–H).

Taken together, the LeX epitopes and their potential carriers were all found in the vicinity of Nestin-positive neural stem cells. Nevertheless, the exact localization of the different molecules was not identical. The rosette lumen, the cell surface, the outer border of the rosettes and the surroundings were labeled in individual patterns. In addition to the LeX motif in different molecular contexts, we were interested in the expression of the neural stem cell-related DSD-1 epitope.

### 3.4. Expression of the DSD-1 Epitope in Cerebral Organoids

The DSD-1 chondroitin sulfate epitope is recognized by the mAb 473HD, which we combined with stainings for the neural stem cell marker Nestin (Figure 6). The DSD-1 epitope was highly enriched at the outer border of neural rosette structures and in the area adjacent to the rosettes after 18 and 53 DIV (Figure 6A,B and Figure A4E). A broad overlap with Nestin-positive cells could be observed outside the rosettes.

After 81 DIV, Nestin as well as the DSD-1 epitope were expressed in a relatively diffuse and in part overlapping pattern (Figure 6C and Figure A4F).

### 3.5. Overview: Neural Stem Cell-Related Carbohydrates in Human Cerebral Organoids

The main findings of the immunohistochemical analysis for the LewisX (mAb 487^LeX^ and mAb 5750^LeX^) and DSD-1 (mAb 473HD) carbohydrate structures and for the potential carrier molecules PTPRZ1 and TNC are summarized in Figure 7. The expression patterns of the different cell type-specific markers are depicted as references.

## 4. Discussion

Before we analyzed the expression of the different glycoepitopes, we started by characterizing the human cerebral organoids with appropriate markers to confirm the cell identities. Different cell fates could be identified, with a clear spatiotemporal pattern: Proliferating, KI-67-positive cells were abundant in the early organoids and were clearly reduced in number over time (Figure 2 and Figure 3). The PH3 staining revealed that late G2- and M-phase cell nuclei were located at the apical side of the rosettes (Figure 2), indicating that interkinetic nuclear migration as a hallmark and important behavior of neural stem cells in the neural tube was resembled in vitro [43]. This also means that the cells are exposed to the same alternating signals as they are in vivo, a mechanism that is proposed by which the cells might be influenced depending on their nucleus’ position within the cell cycle [44]. We identified neural stem/progenitor cells via their Nestin expression, predominantly in the neural rosettes and in older organoids in a rather diffuse distribution (Figure 5). It cannot be excluded that the relatively strong Nestin expression in the older organoids indicated cells with an incomplete differentiation state, maybe caused by the artificial environment. Neurons, labeled with antibodies against βIII-Tubulin, were found outside the rosette structures, in line with the expectation that postmitotic, differentiating cells migrate basally (Figure A3). We used GFAP as a marker for astrocytes. GFAP-expressing astrocytes were found at later stages, although large areas of the organoids were still free of GFAP signals (Figure A3). GFAP is only one of several astrocyte markers and due to their heterogeneity, not all astrocytes express all markers [45]. Therefore, GFAP-negative astrocytes might be present in the organoids that were not labeled in our analysis. Progenitor cells and cells of the oligodendroglial lineage positive for OLIG2 could not be detected in large amounts and the signals appeared not very specific (Figure A3). This is in line with the results of other groups and represents a limitation of the organoid model [46].

In our glycoepitope analysis, we detected the LeX trisaccharide motif, the DSD-1 chondroitin sulfate epitope, as well as their potential carriers PTPRZ1 and TNC directly at or in the vicinity of the neural tube-resembling rosettes formed by Nestin-positive neural stem cells (Figure 7). Although all the factors were related to the neural rosette areas, a closer look revealed that the exact localization of the different molecules was not identical. In principle, the rosette lumen, the surface of the radially shaped cells, the outer border of the rosettes and the surrounding tissue were labeled in individual patterns.

In relatively early organoids, 487^LeX^ signals were detectable at the lumen and at the outer border of the neural rosettes, whereas prominent 5750^LeX^ signals were found in the lumen of the rosettes (Figure 4). PTPRZ1 was found on the cells of the rosettes and TNC was mainly labeled at the rosette borders. In a single-cell RNA-seq approach, TNC has been found to be enriched in the astrocyte population and PTPRZ1 was highly expressed in the “forebrain” cluster of cells in cerebral organoids [47]. The DSD-1 epitope was highly enriched at the outer border of neural rosette structures and in the area adjacent to rosettes (Figure 6). In older organoids, when neural rosettes had disappeared, the staining patterns were more diffuse and harder to distinguish due to a less defined cytoarchitecture. The very distinct expression patterns suggest that these carbohydrates mediate specific functions in the developing human nervous system. The differences observed between the LeX antibodies 487^LeX^ and 5750^LeX^ had not been detected in a 2D adherent culture with neural rosettes before [26]. This might be due to technical issues, as the cells in the adherent culture were not that well-organized compared to the organoid approach. However, it is also possible that the more in vivo-like conditions in the 3D system compared to the 2D culture provide an environment where the cells could establish a more sophisticated morphology, reflected by a different ECM expression pattern. The individual enrichment of ECM at the apical or the basal side of the radially shaped neural stem cells reflects the polarity of these cells, a characteristic that is replicated in vitro [48]. A differential expression of the LeX epitopes is not surprising per se. For rodent neural stem cells, subpopulations of cells have been labeled that exclusively expressed the epitope of mAb 487^LeX^ or 5750^LeX^ and the staining patterns seen in our organoids closely resemble the expression patterns in the mouse spinal cord at embryonic day 13.5 [9,49].

We focused our analysis on TNC and PTPRZ1 as potential LeX carriers. TNC is an important component of the embryonic and adult neural stem cell niche [16,17]. PTPRZ1 isoforms are also expressed in the immature nervous system, where they regulate neural stem cell behavior [11,19]. Interestingly, both molecules are expressed by SOX2-positive outer radial glia (synonym: basal radial glia) in the gyrified human neocortex, a specialized type of neural stem cell that is responsible for the generation of a huge number of neurons that in the end lead to bending and folding of the tissue [50]. This emphasizes the importance of human in vitro models to complement approaches with animal models.

The expression patterns we observed for the glycoepitopes overlapped with their potential carriers, which is plausible and suggests that indeed TNC and PTPRZ1 carry the carbohydrates in the human cerebral organoid model. Of course, it is possible that additional LeX carriers are expressed in the organoids, for example, the Low density receptor-related protein 1 (LRP1) and others. PTPRZ1 is a LeX carrier, but at the same time, the only confirmed carrier of the DSD-1 epitope [19,51]. Therefore, we do not expect the expression of the DSD-1 epitope on other molecules.

The question is how glycoepitope expression can eventually result in cellular responses. As a secreted molecule, TNC depends on indirect effects via receptors or its interaction with other molecules, such as growth factors [50,52]. Transmembrane PTPRZ1 isoforms possess two intracellular tyrosine phosphatase domains, of which only one is catalytically active [53]. It is apparent that carbohydrate-mediated interactions on the cell surface, for example of PTPRZ1 with its ligand Pleiotrophin, induce intracellular signaling cascades [54]. These cascades comprise β-Catenin, which is part of Wnt signaling, and Fyn tyrosine kinase [55,56]. Wnt and Fyn are important regulators of neural stem cells [57,58]. In turn, this means that intracellular signaling can be regulated by the carbohydrates on the extracellular side. In contrast, Fibroblast growth factor 2 (FGF2) binding relies on the core protein of PTPRZ1 [59]. This shows how individual the molecular mechanisms behind different interactions are.

A general challenge regarding the analysis of secreted molecules is the identification of the producing cell within a tissue. It cannot be excluded that other cells express factors on their surface that bind and thereby enrich a molecule that originates from another cell. For carbohydrates, there is no mRNA that could be detected via in situ hybridization. Instead, only an indirect analysis is possible, by detecting the enzymes that are involved in glycoepitope synthesis. For example, Fucosyl transferase 9 is involved in the synthesis of the LeX motif and different sulfotransferases are essential for the formation of the DSD-1 epitope with its specific sulfation pattern [9,60].

With the results of the present study in mind, open questions remain that should be addressed in future studies. As the expression patterns we observed in the organoids suggest a distinct effect of the carbohydrate structures on the cells’ behavior, blocking experiments would be interesting where the mAbs 487^LeX^, 5750^LeX^, and 473HD are added to the cerebral organoid culture. This would allow a very specific manipulation compared to approaches based on the degradation of the carbohydrates or blocking sulfation in general. A limitation of the classical cerebral organoids is the reduced complexity with regard to non-neural cell types. For example, the neural stem cell niche in vivo also includes signals from the vasculature or microglia [61,62]. Protocols without SMAD inhibitors in the medium allow the generation of innate microglia in cerebral organoids [63]. Assembloids are fused organoids that allow modeling more complex situations like the interaction of cells with different regional identities or the formation of vasculature in organoids [64,65,66,67].

## 5. Conclusions

The differential expression patterns of the LeX and DSD-1 motifs in human cerebral organoids, associated with neural stem cells, indicate a very specific role of carbohydrate modifications in the early steps of human central nervous system development.

## Figures and Tables

**Figure 1 cells-11-00760-f001:**
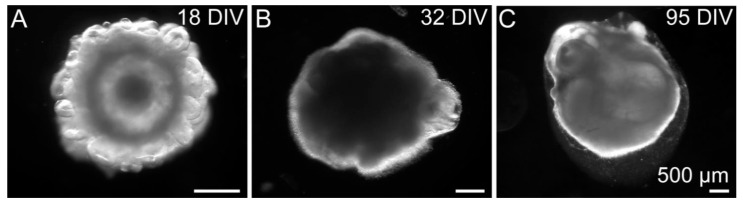
Morphology of human cerebral organoids over time. (**A**) After neural induction, small protrusions were visible on the surface of the early cerebral organoid. (**B**,**C**) The surface became smoother, as shown after 32 DIV and 95 DIV. Here, only a few, larger protrusions could be observed.

**Figure 2 cells-11-00760-f002:**
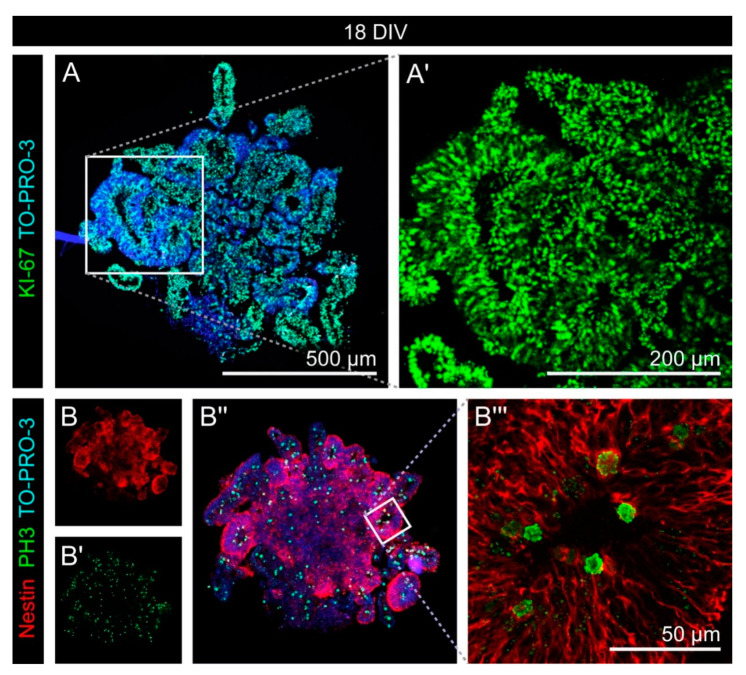
Proliferation in early cerebral organoids. (**A**,**A′**) KI-67 (green), which labels all proliferating cells, was expressed by the vast majority of all cells labeled with nuclear TO-PRO-3 staining (blue). (**B**–**B‴**) Phospho-Histone H3 (green), a marker restricted to late G2- and M-phase cells, was expressed by a minority of cells. Within the neural rosette-like structures formed by Nestin-positive cells (red), PH3 signals were primarily found at the apical side, near the lumen.

**Figure 3 cells-11-00760-f003:**
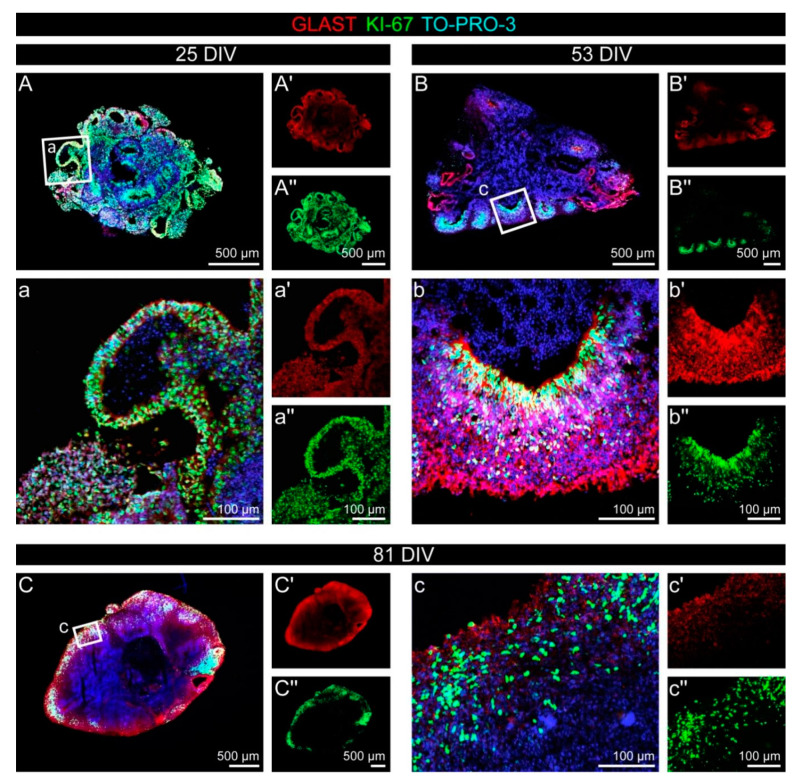
Proliferation in cerebral organoids over time. (**A**–**c″**) The ratio of proliferating cells was reduced over time, as the comparison of KI-67 (green) in an early cerebral organoid after 25 DIV (**A**–**a″**) with the situation after 53 DIV (**B**–**b″**) or even 81 DIV (**C**–**c″**) shows. GLAST (red), a marker for the radial glia type of neural stem cells, was strongly expressed in the neural rosette structures in the early organoids, whereas the signal appeared more diffuse after 81 DIV.

**Figure 4 cells-11-00760-f004:**
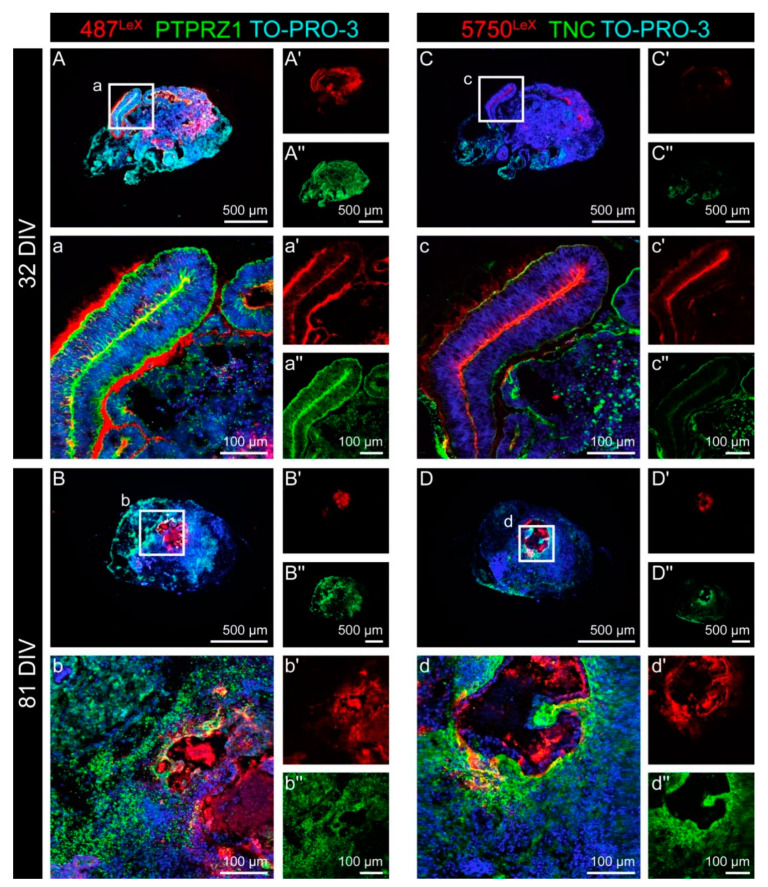
Double staining for the LewisX (LeX) motif and for the potential LeX carrier molecules PTPRZ1 and TNC in cerebral organoids. (**A**–**b″**) The monoclonal antibody (mAb) 487^LeX^ binds to terminal LeX motifs (red) and was clearly enriched in the lumen and at the outer border of neural rosette structures after 32 DIV. PTPRZ1 (green) showed a similar, but not identical staining pattern. The strongest signals were found on the cells within the rosette structures. After 81 DIV, the signals of 487^LeX^ and for PTPRZ1 appeared more diffuse and showed no clear co-localization at this stage. (**C**–**d″**) mAb 5750^LeX^ (red) was used to detect internal LeX motif repeats. The signals were located near the lumen of neural rosettes after 32 DIV, whereas the outer borders of such areas were only weakly stained. TNC (green) was mainly labeled at the rosette borders. After 81 DIV, signals for mAb 5750^LeX^ and for TNC could still be detected in the organoid. Here, regions with intensely labeled cells on the one hand and regions with only faint signals, on the other hand, were observed.

**Figure 5 cells-11-00760-f005:**
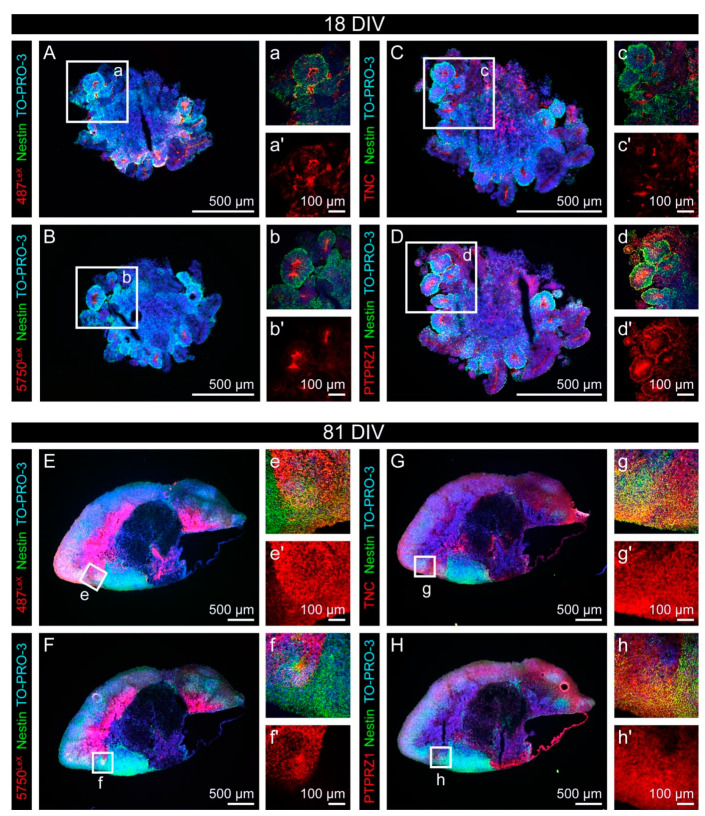
Expression of the LewisX (LeX) glycoepitope and of the potential LeX carriers TNC and PTPRZ1 with regard to Nestin-positive progenitor cells in cerebral organoids. (**A**–**d′**) After 18 DIV, Nestin signals were most prominent in the neural rosettes (green). Signals for 487^LeX^ (red) were detectable at the lumen and at the outer border of neural rosettes. In contrast, 5750^LeX^ signals (red) were most intense in the lumen of these structures. TNC (red) was also detectable on cells of the neural rosettes, although it appeared weaker compared to PTPRZ1 (red). (**E**–**h′**) After 81 DIV, the rosette-like structures disappeared and the staining patterns of the molecules changed accordingly. Nestin-positive cells were present throughout the organoid. Signals of mAb 487^LeX^, mAb 5750^LeX^, for TNC and for PTPRZ1 were detectable, without a clear expression pattern.

**Figure 6 cells-11-00760-f006:**
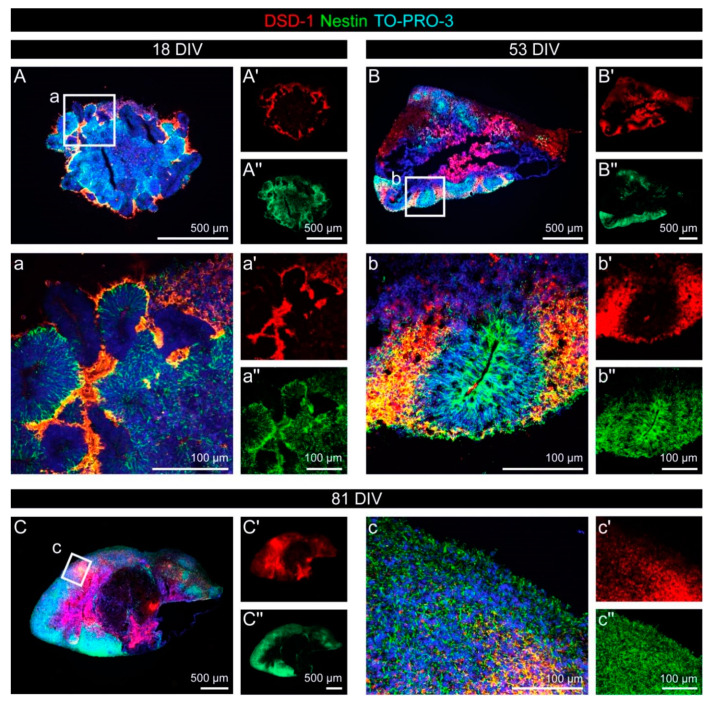
Expression of the neural stem cell-related DSD-1 chondroitin sulfate epitope with regard to the neural stem cell marker Nestin in cerebral organoids. (**A**–**b″**) After 18 and 53 DIV, the DSD-1 epitope (red) was highly enriched at the outer border of neural rosette structures and in the area adjacent to the rosettes. A broad overlap with Nestin-positive (green) cells could be observed. (**C**–**c″**) Later, after 81 DIV, Nestin, as well as the DSD-1 epitope, were labeled in a relatively diffuse and in part overlapping pattern.

**Figure 7 cells-11-00760-f007:**
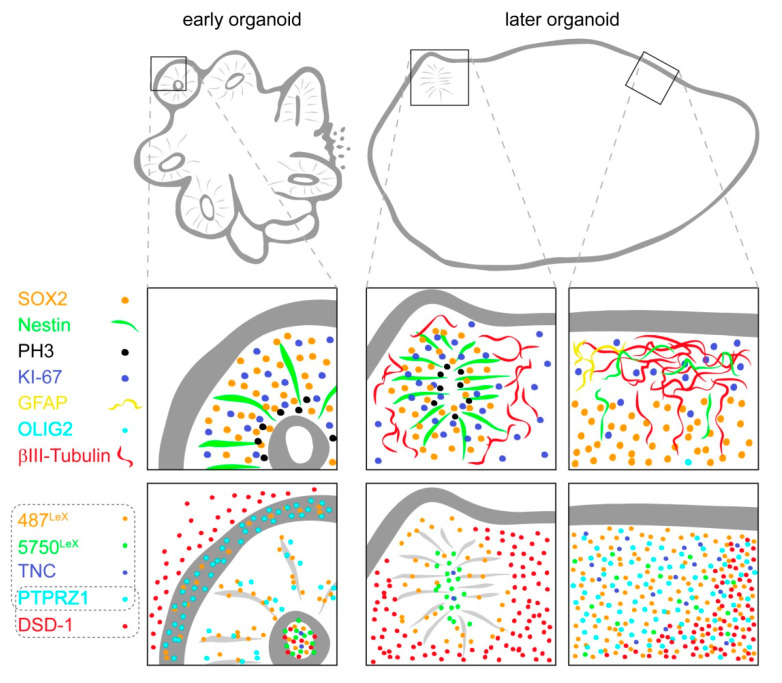
Schematic representation of neural stem cell-associated glycoepitopes and associated extracellular matrix molecules in human cerebral organoids over time. Important cell type-specific markers are indicated as reference.

**Table 1 cells-11-00760-t001:** Primary antibodies used for immunohistochemistry.

Antibody	Species	Source	Cat#	Lot	Dilution (IHC)	RRID/Reference
473HD (DSD-1 epitope)	Rat IgM	Faissner et al. (1994)	N/A	12/06	1:350	[40]
487^LeX^ (LewisX epitope)	Rat IgM	Streit et al. (1990)	N/A	9/08	1:350	[41]
5750^LeX^ (LewisX epitope)	Rat IgM	Hennen et al. (2011)	N/A	2/08	1:50	[9]
GFAP	Mouse IgG	Sigma-Aldrich	G3893	038M4864V	1:300	RRID:AB_477010
GLAST (EAAT1)	Rabbit IgG	Cell Signaling Technology	5684	0001	1:100	RRID:AB_10695722
KAF12 (Tenascin C)	Rabbit polyclonal	Faissner and Kruse (1990)	N/A	7/96	1:300	[42]
KAF13 (PTPRZ1)	Rabbit polyclonal	Faissner et al. (1994)	N/A	N/A	1:300	[40]
KI-67	Mouse IgM	Sigma-Aldrich	P6834	034144776	1:50	RRID:AB_261141
Nestin	Mouse IgG	Santa Cruz Biotechnology	sc-23927	A2914	1:250	RRID:AB_627994
OLIG2	Rabbit polyclonal	Millipore	AB9610	3071572	1:400	RRID:AB_570666
Phospho-Histone H3	Rabbit polyclonal	Millipore	06-570	2825969	1:300	RRID:AB_310177
SOX2	Rabbit polyclonal	Millipore	AB5603	3108482	1:100	RRID:AB_2286686
βIII-Tubulin	Mouse IgG	Sigma-Aldrich	T8660	097M4835V	1:250	RRID:AB_477590

**Table 2 cells-11-00760-t002:** Secondary antibodies used for immunohistochemistry.

Antibody; Conjugation	Species	Cat#	Lot	Dilution (IHC)	Source/RRID
Anti-rabbit IgG (H + L); Alexa Fluor^®^ 488	Goat	111-545-045	135978	1:300	Jackson ImmunoResearch, Ely, UK, RRID:AB_2338049
Anti-rabbit IgG (H + L); Cy™3	Goat	111-165-045	125365	1:300	Jackson ImmunoResearch, Ely, UK, RRID:AB_2338003
Anti-mouse IgG + IgM (H + L); Alexa Fluor^®^ 488	Goat	115-545-044	130731	1:300	Jackson ImmunoResearch, Ely, UK, RRID:AB_2338844
Anti-mouse IgG + IgM (H + L); Cy™3	Goat	115-165-068	136057	1:300	Jackson ImmunoResearch, Ely, UK, RRID:AB_2338686
Anti-rat IgM (µ-chain); Cy™3	Goat	112-165-075	128380	1:300	Jackson ImmunoResearch, Ely, UK, RRID:AB_2338249

## Data Availability

The data presented in this study are available in the main article and in the Appendix A, Appendix B and Appendix C.

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
