# Peer review of "Cerebral Organoids Maintain the Expression of Neural Stem Cell-Associated Glycoepitopes and Extracellular Matrix"

_cells, 2022, doi:10.3390/cells11050760_

Round 1
Reviewer 1 Report
Brief summary:
The authors present a detail characterization of the expression pattern of selected markers in developping human cerebral organoids, focusing on the LewisX motif and DSD-1. The mapping of the selected markers is performed using immunohistochemistry and fluorescence imaging, showing the differential localization of each marker with respect to time. The topic is introduced in a detailed and approachable manner, however a few issues should be addressed.
General comments:
- The introduction presents a clear and concise overview of the topic, but omits to mention the rationale for the selection of PTPRZ1 and TNC as carriers. This information is listed later in the results and discussion sections, but mentioning it in the introduction would also be beneficial.
- After lines 64-67, please cite appropriate review articles on murine/human brain differences. E.g. Miller, Horvath and Geschwind (doi: 10.1073/pnas.0914257107).
- In the Materials and Methods section, the word "exchange" is used to describe the changing or replacement of medium. It would be appropriate to use "change" or "replace", as it is later in the text.
- In lines 131-132, the term "CO2 incubator" is used. If referring to a cell culture incubator (37C, 5% CO2), it would be preferrable to simply use "incubator" to avoid confusion.
- In the Materials and Methods section, there is no description of the apparatus used for fluorescence imaging. Please provide the model and settings used.
- The results of the manuscript are presented visually as clear and attractive fluorescence micrographs. Given the known variability between organoids, the results would be significantly more compelling if quantified, with the number of organoids/slices analyzed listed for each time point.
- The discussion of organoid challenges and limitations is appreciated (lines 428-430). The citation of Ormel et al. (https://doi.org/10.1038/s41467-018-06684-2) would enrich the topic of microglia in organoids.
Author Response
Point 1: Brief summary: The authors present a detail characterization of the expression pattern of selected markers in developing human cerebral organoids, focusing on the LewisX motif and DSD-1. The mapping of the selected markers is performed using immunohistochemistry and fluorescence imaging, showing the differential localization of each marker with respect to time. The topic is introduced in a detailed and approachable manner, however a few issues should be addressed.
General comments: The introduction presents a clear and concise overview of the topic, but omits to mention the rationale for the selection of PTPRZ1 and TNC as carriers. This information is listed later in the results and discussion sections, but mentioning it in the introduction would also be beneficial.
Response 1: We thank the reviewer for this and the other very helpful comments and now explain the importance of the glycoepitope carriers already in the introduction.
Point 2: After lines 64-67, please cite appropriate review articles on murine/human brain differences. E.g. Miller, Horvath and Geschwind (doi: 10.1073/pnas.0914257107).
Response 2: As suggested, we included the Miller article and other references to underline the need for human tissue models.
Point 3: In the Materials and Methods section, the word "exchange" is used to describe the changing or replacement of medium. It would be appropriate to use "change" or "replace", as it is later in the text.
Response 3: We replaced the term “exchange” as suggested.
Point 4: In lines 131-132, the term "CO2 incubator" is used. If referring to a cell culture incubator (37C, 5% CO2), it would be preferable to simply use "incubator" to avoid confusion.
Response 4: Indeed, we refer to a cell culture incubator. To avoid confusion, we now write “incubator”.
Point 5: In the Materials and Methods section, there is no description of the apparatus used for fluorescence imaging. Please provide the model and settings used.
Response 5: We apologize for omitting these technical details and included information on the microscopes and the settings that were used.
Point 6: The results of the manuscript are presented visually as clear and attractive fluorescence micrographs. Given the known variability between organoids, the results would be significantly more compelling if quantified, with the number of organoids/slices analyzed listed for each time point.
Response 6: Indeed, there is a high variability between organoids. We decided to describe the distribution of the different expression patterns, as it is hard or even impossible to attribute secreted extracellular matrix molecules to a certain cell. We observed the glycoepitope expression patterns in at least three young organoids (up to five weeks after embedding in Matrigel) with the typical neural rosette structure and at least three later organoids (at least six weeks after embedding in Matrigel), respectively. This is now mentioned in the Materials and Methods section.
Point 7: The discussion of organoid challenges and limitations is appreciated (lines 428-430). The citation of Ormel et al. (https://doi.org/10.1038/s41467-018-06684-2) would enrich the topic of microglia in organoids.
Response 7: As microglia is an important player in the brain, we included the study of Ormel et al., who were able to identify microglia cells in cerebral organoids.

Reviewer 2 Report
This paper aims to analyze expression patterns of specific carbohydrates molecules bound to extracellular matrix proteins in human cerebral organoids at different stages of "development". The authors focus on LewisX and DSD-1.
This descriptive article is very clear, the methods are complete and well described, the figures are clear. Figure 7 brings a very nice summary of the findings.
Author Response
Point 1: This paper aims to analyze expression patterns of specific carbohydrates molecules bound to extracellular matrix proteins in human cerebral organoids at different stages of "development". The authors focus on LewisX and DSD-1.
This descriptive article is very clear, the methods are complete and well described, the figures are clear. Figure 7 brings a very nice summary of the findings.
Response 1: We thank the reviewer for the positive feedback on our manuscript and the illustration.

Reviewer 3 Report
This paper is very interesting and helpful for tissue engineering, stem cell, or biology researchers. However, the authors should introduce the 3D cell culture, such as spheroids or organoids, to enhance the cell function or mimic the body environment. The readers must be confused. In addition, not only the microscopic pictures but also numerical value is essential, such as gene or protein expression. By performing statistical analysis, the authors can conclude the study. Moreover, the authors should discuss this study by quoting related research papers. Taken together, major revisions should be made before re-submission. The paper would be re-considered only when all the comments were responded.
Introduction and Discussion
There are many reports on the 3D tissue models for cell function, such as migration or protein expression. Unfortunately, the introduction of this field is too poor. The authors should add some sentences for the description of the fields. To reduce the authors’ burden, I suggest at least these recent papers be added for revision (review and research paper).
Review papers (for concept)
Cancers 2020, 12(10), 2754
Journal of Cell Communication and Signaling volume 5, Article number: 239 (2011)
Research papers
For protein expression: Cell175, 1591–1606
For cell migration: Tissue Eng. Part C Methods 2019, 25, 711–720. https://doi.org/10.1089/ten.tec.2019.0189
For drug resistance: Nature Methods volume 15, pages134–140 (2018)
For cell alternation https://doi.org/10.1093/toxsci/kfx289
The authors should investigate the gene or protein expression of organoids.
I think the cells are dead in the center of organoids due to hypoxia. This size of organoids is too large to permeate the oxygen. The authors should discuss this point.
The self-citation ratio is very high: 16/50
This is not good, so many references should be quoted and introduced.
Author Response
Point 1: This paper is very interesting and helpful for tissue engineering, stem cell, or biology researchers. However, the authors should introduce the 3D cell culture, such as spheroids or organoids, to enhance the cell function or mimic the body environment. The readers must be confused. In addition, not only the microscopic pictures but also numerical value is essential, such as gene or protein expression. By performing statistical analysis, the authors can conclude the study. Moreover, the authors should discuss this study by quoting related research papers. Taken together, major revisions should be made before re-submission. The paper would be re-considered only when all the comments were responded.
Introduction and Discussion
There are many reports on the 3D tissue models for cell function, such as migration or protein expression. Unfortunately, the introduction of this field is too poor. The authors should add some sentences for the description of the fields. To reduce the authors’ burden, I suggest at least these recent papers be added for revision (review and research paper).
Review papers (for concept) Cancers 2020, 12(10), 2754; Journal of Cell Communication and Signaling volume 5, Article number: 239 (2011); Research papers For protein expression: Cell 175, 1591–1606; For cell migration: Tissue Eng. Part C Methods 2019, 25, 711–720. https://doi.org/10.1089/ten.tec.2019.0189; For drug resistance: Nature Methods volume 15, pages 134–140 (2018); For cell alteration: https://doi.org/10.1093/toxsci/kfx289
Response 1: We thank the reviewer for the helpful and very detailed feedback. As suggested, we expanded the introduction with regard to the different aspects that can be analyzed with 3D culture models. To keep the introduction relatively straightforward and focused on the glycoepitopes, we could not go too much into detail.
Point 2: The authors should investigate the gene or protein expression of organoids.
Response 2: This is an important aspect. Due to the individual differences between organoids, also with regard to the number and/or position of the rosette-like structures and the high number of other cells that would be pooled in the lysate, we think that measuring the overall amount of mRNA (for the carriers) or proteins/carbohydrates in the organoids is not the optimal parameter for our research question. Of course, mRNA analysis on the single cell level would be desirable, but is out of the scope of the study. Therefore we focused our analysis on the expression patterns with respect to the cell types and their position in the organoids.
Point 3: I think the cells are dead in the center of organoids due to hypoxia. This size of organoids is too large to permeate the oxygen. The authors should discuss this point.
Response 3: Indeed, the signals in the center of the late organoids are not reliable and the analysis is limited to the outer regions of the organoids. We mention this in the Results part now: “The strong signals in the core of late organoids indicate necrosis due to insufficient oxygen supply in the cell culture and therefore cannot be considered specific.”
Point 4: The self-citation ratio is very high: 16/50
This is not good, so many references should be quoted and introduced.
Response 4: The ratio of self-citation was high, as we tried to keep the manuscript as concise as possible but to introduce the background of the molecules quite detailed at the same time. As described above, we expanded the introduction to provide more information on the 3D culture/organoid context with more references, which reduces the self-citation ratio accordingly.

Round 2
Reviewer 1 Report
The present manuscript has been improved on minor points, however the authors have declined to modify the most important point, which is to add quantifications of the data to support the qualitative observations.
They admit that there is extremely high variability between organoid, yet only make qualitative observations based on three organoids per time point to support their conclusions. I do not consider this strong enough.
Author Response
Point 1: The present manuscript has been improved on minor points, however the authors have declined to modify the most important point, which is to add quantifications of the data to support the qualitative observations.
They admit that there is extremely high variability between organoid, yet only make qualitative observations based on three organoids per time point to support their conclusions. I do not consider this strong enough.
Response 1: We thank the reviewer for the comment. To support our findings, we have now added representative fluorescence intensity profiles for 487LeX, 5750LeX and DSD-1, combined with Nestin and TO-PRO-3 nuclear staining, in early and late cerebral organoids (shown in Figure B4 in the Appendix).
Reviewer 3 Report
The authors have responded to all comments.
Author Response
Point 1: The authors have responded to all comments.
Response 1: We thank the reviewer for checking our revised manuscript.
Round 3
Reviewer 1 Report
I appreciate the authors' receptiveness and amendements, particularly for Figure B4.